# Security and Privacy Issues in Medical Internet of Things: Overview, Countermeasures, Challenges and Future Directions

**Mohamed Elhoseny [1], Navod Neranjan Thilakarathne [2,*], Mohammed I. Alghamdi [3], Rakesh Kumar Mahendran [4], Akber Abid Gardezi [5], Hesiri Weerasinghe [6] and Anuradhi Welhenge [6]**

[1] Faculty of Computers and Information, Mansoura University, Mansoura 35516, Egypt; melhoseny@ieee.org
[2] Department of ICT, Faculty of Technology, University of Colombo, Colombo 00700, Sri Lanka
[3] Department of Computer Science, Al-Baha University, Al-Bahah 1988, Saudi Arabia; mialmushilah@bu.edu.sa
[4] Department of Electronics and Communication Engineering, Vel Tech Multitech Dr. Rangarajan Dr. Sakunthala Engineering College, Tamil Nadu 6000062, India; rakeshkumarmahendran@gmail.com
[5] Department of Computer Science, COMSATS University Islamabad, Islamabad 44000, Pakistan; akber.gardezi@comsats.edu.pk
[6] Faculty of Computing and Technology, University of Kelaniya, Colombo 00700, Sri Lanka; hesiri@kln.ac.lk (H.W.); anuradhiw@kln.ac.lk (A.W.)
**\*** Correspondence: navod.neranjan@ict.cmb.ac.lk

**Abstract:** The rapid development and the expansion of Internet of Things (IoT)-powered technologies have strengthened the way we live and the quality of our lives in many ways by combining Internet and communication technologies through its ubiquitous nature. As a novel technological paradigm, this IoT is being served in many application domains including healthcare, surveillance, manufacturing, industrial automation, smart homes, the military, etc. Medical Internet of Things (MIoT), or the use of IoT in healthcare, is becoming a booming trend towards improving the health and wellbeing of billions of people by offering smooth and seamless medical facilities and by enhancing the services provided by medical practitioners, nurses, pharmaceutical companies, and other related government and non-government organizations. In recent times, this MIoT has gained higher attention for its potential to alleviate the massive burden on global healthcare, which has been caused by the rise of chronic diseases, the aging population, and emergency situations such as the recent COVID-19 global pandemic, where many government and non-government medical resources were challenged, owing to the rising demand for medical resources. It is evident that with this recent growing demand for MIoT, the associated technologies and its interconnected, heterogeneous nature adds new concerns as it becomes accessible to confidential patient data, often without patient or the medical staff consciousness, as the security and privacy of MIoT devices and technologies are often overlooked and undermined by relevant stakeholders. Hence, the growing security breaches that target the MIoT in healthcare are making the security and privacy of Medical IoT a crucial topic that is worth scrutinizing. In this study, we examined the current state of security and privacy of the MIoT, which has become of utmost concern among many security experts and researchers due to its rapid demand in recent times. Nevertheless, pertaining to the current state of security and privacy, we also examine and discuss a number of attack use cases, countermeasures and solutions, recent challenges, and anticipated future directions where further attention is required through this study.

**Keywords:** security; privacy; IoT; medical internet of things; smart health

## 1. Introduction

Technology integration is becoming an integral part of our daily life as a result of the technological advancement of various technologies [1]. This results in less manual work and aids in ubiquitously interconnecting everyone, and IoT plays a major role, offering

smooth and seamless ubiquitous services for everyone [2,3]. In general, the IoT refers to the networking of physical devices that are smart and interconnected [4] and comprises sensors, software, and network connectivity that enables it to collect and exchange data [5,6]. Currently, the IoT is shaping and transforming both the business and consumer worlds, finding its way into every global business and consumer domain. Apart from this, it is also being delivered in many other domains, including healthcare, smart cities, agriculture, the military, and so on [4–8]. Hence, the IoT may significantly enhance the way people interact with the world. Based on recent reports, the IoT market size was valued at USD 761.4 billion in 2020 and is projected to reach USD 1386.06 billion by 2026, which signifies its importance as a dominant technological paradigm towards improving the well-being of billions of people all around the world [9–12].

When it comes to the IoT in healthcare, or what is well known as MIoT, it refers to a wide variety of IoT devices whose main purpose is to facilitate and aid in fundamental patient care [7–11]. the global IoT in healthcare market size is USD 71.84 billion in 2020 and the market is projected to grow from USD 89.07 billion in 2021 to USD 446.52 billion by 2028 [9–12]. As of now, healthcare providers are utilizing various MIoT based applications and services for patient treatment, disease management, medical diagnosis, to improve patient care, and lower the costs of care [4–8], where they are capable of collecting various information such as vital body parameters from patients and monitor pathological details by implantable medical sensors or small wearable sensors that are worn by the patient [11–18]. With the aid of MIoT devices, patient condition can be monitored remotely and in real-time, and the captured data can then be analyzed and transmitted to the cloud data storage or the medical data centers for further processing and storage before offering services to various stakeholders such as physicians and other related medical staff, caregivers, and insurance service providers [18,19]. In general, MIoT applications include solutions that are designed for remote health monitoring, emergency patient care, healthcare management, the monitoring of elderly patients, clinical decision support systems, wireless capsule endoscopy, and so on [4,7,8,20–24]. Nevertheless, it is also evident that now the IoT has revolutionized healthcare organizations to expand their services to in-home patients where they can monitor, track, and treat the conditions of patients remotely while they are engaging with their daily activities. A typical MIoT system can be compartmentalized into several components [24–31]. For better understanding, an overview of a general MIoT system in healthcare is depicted in Figure 1.

In general, a typical MIoT healthcare system comprises of a series of medical IoT devices that are embedded with various kinds of intelligent sensors that can perceive their surroundings [31,32]. The collected medical data can be processed either by a smart device itself or in the cloud, where a wireless medium is used to transfer the collected health information to the relevant stakeholders, which helps them to make decisions about the patient's condition [10,22,32–35]. In addition, such smart devices can be further connected to worldwide information networks for convenient and on-demand access.

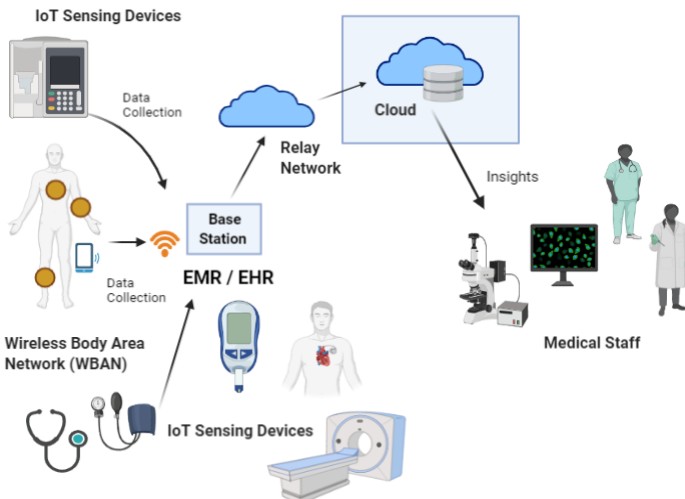

**Figure 1.** Overview of a general MIoT system in healthcare.

The integration and the connectivity of physical things in the MIoT environment to the Internet means the possibility of remote access to the devices, which are made for the monitoring, analysis, forecasting, and storage of vital medical data [36,37]. On the other hand, these integrated IoT technologies introduce various vulnerabilities, owing to the rapidly evolving IoT threat landscape, where intruders can exploit and gain access to the MIoT network to further exploit the entire medical network/environment. This ultimately leads to situations where the security and privacy of the devices and patients are at risk. As the volume of data that is handled and generated by MIoT devices grows exponentially, it will eventually lead to the greater exposure of confidential medical data, which necessitates further study on the matter. Hence, the security and privacy of the data obtained from MIoT devices, which are either stored in the cloud or in remote servers or obtained during the transmission to the cloud or remote servers, are becoming a major unresolved concern in healthcare, where less attention is paid by the industry and the academic community [10,17]. Moreover, based on recent statistics [30–35], the healthcare sector leads in terms of the sectors that have been breached by cyber-attacks in recent times, as depicted in Figure 2 [15–20].

**Number of Records Breached by Industry**

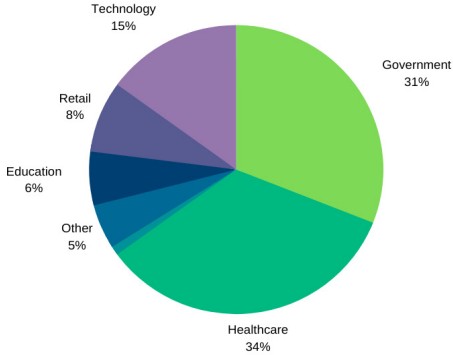

**Figure 2.** Latest statistics of the number of records breached by industry.

With the exponential growth of Internet-connected MIoT devices, confidential patient information is exposed to outside parties that can be accessible in numerous ways. For an instance, an intruder can eavesdrop through the wireless communication network and sniff for data that is being exchanged over the wireless network to access confidential patient data. In worst-case scenarios, intruders can remotely access the control unit of the medical device and can then control the device, jeopardizing the lives of patients [5,37]. Moreover, it would be a huge threat to the patient's privacy if a passive network observer could infer confidential patient information from the network traffic, especially when then inferred information can be used for abusive purposes following the attack [38,39]. It is evident that the lack of adequate knowledge about MIoT security among the end-users and the relevant stakeholders (e.g., medical staff, patients, caregivers) may also exacerbate vulnerabilities and may encourage attackers to further exploiting MIoT technologies, ultimately endangering the lives of patients in most cases [5,8,10,12]. Not only that, but in case of any cyber-attack, the biggest concerns or threats for healthcare would be data leakage or information loss, eventually resulting patient data being compromised, as depicted in Figure 3. Nevertheless, recent trends indicate that with the ever-increasing cyber-attacks that target healthcare, the healthcare IoT security market is expected to undergo rapid growth by the year 2025, with a total revenue of USD 100 billion, which also justifies our efforts to examine the current state of security and privacy of the MIoT through this study [38–41].

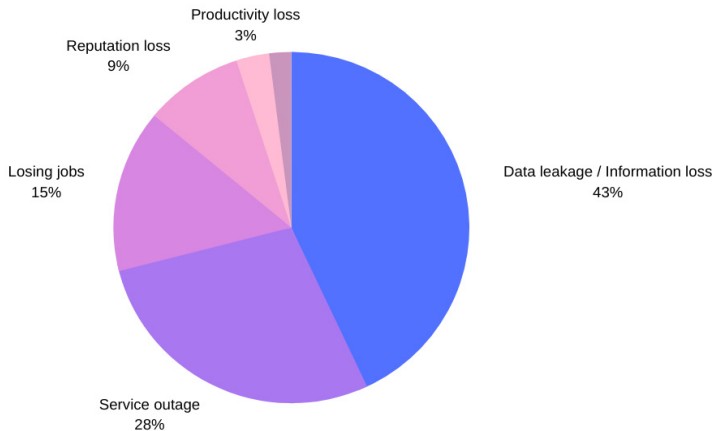

**Figure 3.** Biggest concerns in the healthcare sector if attacked.

On the other hand, in November of 2019, the entire world woke up to a deadly virus outbreak called COVID-19 that quickly spread to countries worldwide, posing worldwide chaos. Up to the moment where we are writing this, the virus has caused by worldwide population to decrease by almost 4 million people, posing huge doubt about the next phases of pandemic management when considering the current level of available medical resources. Global lockdowns have already been imposed, including country-wide and state-wide lockdowns, to contain the spread of the virus. In order to control and contain the spread of the virus, healthcare organizations, with the help of governments, introduced a variety of MIoT-based technologies to track and treat patients remotely, owing to the contagious nature of the virus and to reduce the strain on the medical facilities. As a

result, the demand for MIoT devices and technologies have also grown during this pandemic season, has also led to a boost in MIoT security attacks, according to recent studies [40–45], which further motivated us towards compiling this study.

### 1.1. Motivation of Study

There is no question that a greater guarantee for the health and wellbeing of individuals is offered by the use of MIoT in healthcare, where it also puts a lot of strain on the security and privacy aspects of an MIoT-powered healthcare environment, with the lives of patients being endangered [45–47] if no countermeasures are taken. On the other hand, the usage of MIoT in healthcare is proliferating at a rapid phase, owing to rapid demand over recent years, which makes it impossible to address all security and privacy concerns in a timely manner, with many researchers and vendors currently working towards strengthening the security and privacy aspects of the MIoT ecosystem. Nevertheless, the MIoT concept itself is a novel concept where research activities are still in their early stages in terms of security and privacy. Thus, our motivation behind this study was to understand and collate the current level of knowledge pertaining to the security and privacy aspects of MIoT and to provide opportunities to conduct further research in this area that would be highly beneficial for researchers, academics, and vendors who are interested in the security and privacy aspects of MIoT.

### 1.2. Research Problems and Contribution

There have been rapid contributions in the area of MIoT towards proposing novel solutions for patient condition monitoring, disease diagnosis, and pandemic management, with many research studies and surveys being provided on the topic in general. When it comes to the security and privacy aspects, a few surveys have also been conducted on the topic in general, where it is not able to provide any significant knowledge to conduct future research and to devise sound security solutions towards improving the security and privacy of the pervasive MIoT environment. Hence, in order to address this research gap, this study provides an in-depth review of the security and privacy aspects of MioT, highlighting its ecosystem, key contributions, latest trends, countermeasures and solutions, challenges, and future directions. The following is a summary of our contributions:

1. We provide adequate knowledge about the underlying MIoT ecosystem and highlight its architecture, the key layers that the architecture itself is made out of, and the devices and technologies used in each layer.
2. We provide a discussion about the security and privacy requirements of the MIoT and highlight the latest trends to provide a better understanding of what is happening now.
3. We classify security and privacy attacks in terms of the MIoT layered architecture and suggest countermeasures and solutions to prevent these attacks in terms of the layered architecture.
4. We provide a brief comparison of the existing literature, highlighting its key contributions and limitations to justify our work.
5. In terms of the security and privacy of the MIoT, we highlight key challenges based on the layered architecture and also provide future directions as well.

### 1.3. Outline of Study

In order to provide a comprehensive review, as outlined in Section 1.2, the rest of the paper is organized as follows: In Section 2, we discuss the architecture of the MIoT, including the devices and technologies that have been employed thus far. In Section 3, we discuss the security and privacy requirements of the MIoT, highlighting why it has become an appealing target, including the latest trends. In Section 4, we elaborate on several

attack scenarios to better understand attacks on the MIoT based on its layered architecture. Related work and contributions made by others are discussed and compared in Section 5. In Section 6, we describe countermeasures and solutions for solving current security and privacy problems in terms of the MIoT layered architecture. Challenges and future directions are presented in Section 7. Finally, the conclusions are presented in Section 8.

## 2. The Architecture of MIoT

As the main objective of this research study is to understand and review the current state of security and privacy aspects pertaining to the MIoT, it first is essential to have an understanding of the architecture and the devices and the technologies that are employed, as these creates a foundation towards better understanding the various security and privacy issues and their impact. Hence, in this second section, we mainly discuss the architecture of the MIoT and the devices employed in each layer in the architecture. As shown in Figure 4, the architecture of the MIoT can be seen as an abstraction of three hierarchical layers [11,13,16,24,42–47]. That is, the:

1. Perception layer.
2. Network layer.
3. Application layer.

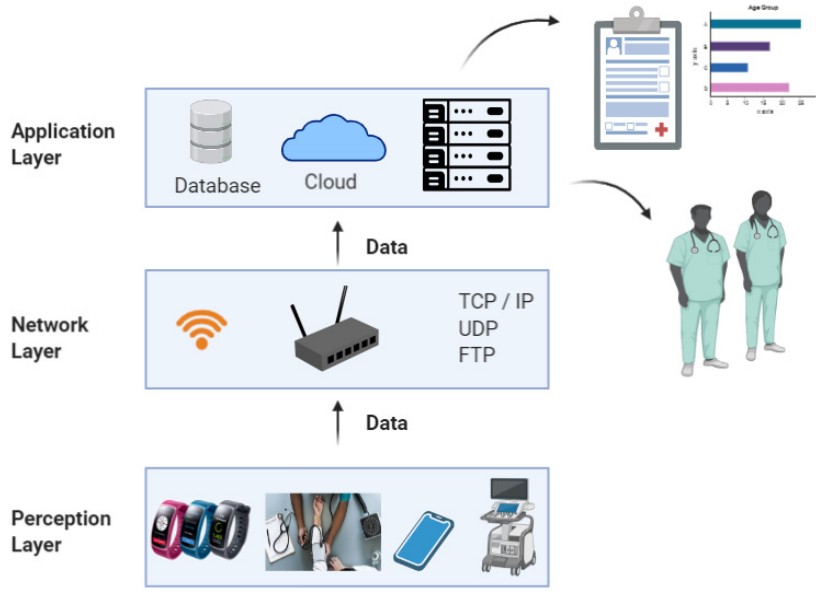

**Figure 4.** The three-layer architecture of the MIoT.

According to the three-layered architecture, the bottommost layer is the perception layer, which is responsible for gathering the medical data from the physical MIoT devices, such as wearable's, smart blood glucose meters, ECG monitoring devices, and so on. Then, the network layer mainly consists of wireless, wired, and middleware systems, which facilitate the smooth delivery of gathered medical data, towards the destination. With the aid of underlying technological platforms, the network layer first processes and communicates the acquired data from the perception layer to the application layer, which is the topmost layer in the architecture. The application layer comprises medical data repositories to offer tailored and personalized medical services and to address the needs of endusers, who are patients, medical professionals, caregivers, and insurance companies [11,17]. An important fact is that the underlying technologies used by each of these layers

are different from one another. Altogether, the MIoT devices and integrated technologies are used to provide a variety of services, each with its own requirements and limitations [13,17].

On the other hand, when it comes to effective service provisioning in MIoT, various protocols are being used, which hold a key place towards improving data transmission efficiency across the three layers and saving energy consumption during data transmission, while also providing security and privacy. The protocols used in each of these layers have their own purpose and characteristics, such as protocols for data transmission, which have connectivity between the MIoT sensors and gateways, and route where they should be evaluated based on the MIoT application type. In Table 1, we highlight the main protocols used in the application and network layers in MIoT solutions, and apportioned them according to the ISO/OSI model in order to generate better understanding.

**Table 1.** Main protocols used in MIoT solutions.

| Network Layer | Application Layer |
|---|---|
| For network layer addressing IPv4/IPv6 protocols are used.<br>For routing RPL, CARP, and CORPL protocols are used.<br>The rest of the protocols used in network layer includes TCP, UDP, 6LoWPAN, IEEE 802.15.4 (e.g.: ZigBee) IEEE 802.15.1 (Bluetooth) LPWAN (e.g.: LoRaWAN) RFID, NFC, IEEE 802.11 (Wi-Fi). | MQTT, CoAP, DSS, AMQP, HTTP, HTTPS, TLS |

*2.1. Classification of Devices Based on the Architecture*

The intention of this subsection is to provide readers a brief overview of the devices and technologies that are employed in each layer based on the three-layer architecture for the MIoT, as previously mentioned, in order to have an exact idea as to the security and privacy issues pertaining to the whole pervasive MIoT ecosystem [40–51].

2.1.1. Devices Employed in the Perception Layer

The perception layer is responsible for accumulating vital body parameters from the patients (e.g., body temperature, blood pressure range, blood oxygen level, heart rate, and blood glucose level, etc.) using physical MIoT devices. The accumulated data are then transferred to the network layer for transportation to the destination. Based on the state of the art, devices in this layer can further be categorized into four categories [13,26,33]:

1. Patient monitoring devices.
2. Remote wellness and chronic disease monitoring devices.
3. Real-time location service (RTL) devices.
4. Facility monitoring devices.

Based on this categorization, further examples are provided in the following subsection for better understanding.

**Patient monitoring devices**

The following Table 2, highlights examples for these patient monitoring devices.

**Table 2.** Patient monitoring devices.

| Device Category | Examples |
| --- | --- |
| Clinical monitors | Heart rate monitors, ventilators, pulse oximetry monitors, electrocardiogram monitors, capnography monitors, depth of consciousness monitors |
| Medical devices | Ventilators, medical imaging devices (e.g., X-rays, computerized tomography (CT) scanners, and magnetic resonance imaging (MRI)), infusion pumps, incubators, smart medical devices, telemetry devices, smart stethoscopes |
| Virtual care devices | Remote ICU telemetry |
| Devices in smart patient rooms | Fall detection monitors, smart beds, personal hygiene monitors |

**Remote wellness and chronic disease monitoring devices**

The following Table 3, highlights examples for these remote wellness and chronic disease monitoring devices.

**Table 3.** Remote wellness and chronic disease monitoring devices.

| Device Category | Examples |
| --- | --- |
| Wearables | Wristbands, bio-energy patches, smart watches |
| Implantable devices | Pacemakers, defibrillators, neurostimulators, respiratory rate sensors, muscle activity sensors, swallowable camera capsules, embedded cardiac devices |
| Remote clinical monitors | Pulse oximeter monitors, ECG monitors, glucometers, fall detection monitors |

**Real-time location service (RTLs) devices**

The following Table 4, highlights examples for these RTL devices used in MIoT based healthcare environment.

**Table 4.** Real-time Location Service (RTLs) devices.

| Device Category | Examples |
| --- | --- |
| Devices for tracking employees | For tracking nursing staff, ancillary staff, physicians |
| Devices for tracking patients | Infant abduction, wandering systems, rehabilitation systems |
| Devices for tracking visitors | Way finding and digital signage |
| Devices for tracking assets | For wheelchairs, infusion pumps, smart cabinets, medication carts |

**Facility monitoring devices**

The following Table 5, highlights examples for these facility monitoring devices in MIoT based healthcare environment.

**Table 5.** Facility monitoring devices.

| Device Category | Examples |
| --- | --- |
| Devices used for environmental controls | Lighting (Daylight sensors), room control, humidity monitoring, water quality monitoring, HVAC |
| Devices used for building management | Elevators, power monitoring and power distribution |
| Devices used for security monitoring | Door locks and entry systems, fire alarms, video surveillance systems |

2.1.2. Devices Employed in the Network Layer

The network layer has the responsibility of distributing content and routing the content to the destination as well as network addressing [13,46–49]. The devices used in this layer are as follows:

- Wired/Wireless media: It is evident that MIoT devices often use wired or wireless networks to connect to the end-user or the gateway [5,14]. In addition, MIoT devices can be connected to Wireless Sensor Networks (WSNs), which use a traditional Wi-Fi network or low-powered wireless personal area network (6LoWPAN). On the other hand, most devices that use wired connections are stationary (e.g., medical imaging devices) [26].
- Radio communication media: Some low-powered MIoT mobile devices use radio communication media such as Bluetooth, RFID, Bluetooth Low Energy (BLE), NFC, and all sorts of cellular communication networks to connect with each node and with end-users and the gateways. Many of the wearable medical IoT devices use BLE for short-range communication. Cellular networks (e.g., 2G, 3G, 4G, 5G) are used for long-range communication [14,25].

2.1.3. Devices Employed in the Application Layer

The application layer bridges physical MIoT devices and the end-users. When the integrated data from the perception layer come to the application layer via the network layer, the collated data are further processed into meaningful information and are saved in repositories in the cloud or dedicated servers in order to provide services as per stakeholder needs. It is evident that most device manufacturers switch their applications towards being hosted in the cloud, owing to the rapid elasticity, convenience, and high scalability that the cloud offers, as opposed to offering services through dedicated servers [13,14].

**3. Security and Privacy Requirement of MIoT**

It is no doubt that MIoT security and privacy play a vital role in modern ubiquitous healthcare [12,16,17], as most healthcare organizations do not devote the adequate time and necessary resources to safeguard security and privacy. A typical MIoT system is a complex ecosystem comprising heterogeneous components (e.g., medical information systems, gateways, cloud services, databases, and smart devices) that can leverage healthcare into the next level [7]. These devices pertaining to the ecosystem generate vast quantities of highly sensitive, real-time, and diversified data [14,35–48], which need to be protected by all means. The need for various strategies to ensure sufficient security and privacy is indicated by the fact that personal medical data are collected and distributed via public or private networks that are insecure most of the time. Thus, when developing robust and secure medical IoT systems, the following requirements should be considered and satisfied [4,5,7,9,11,17,19,20,28,42,50–57]:

- Confidentiality: Confidentially ensures that only authorized personnel have access to the medical data while hindering access for unauthorized personnel [11,42].

- Data integrity: Ensures that an adversary cannot attempt to alter or tamper medical data during transmission or storage [11,29].
- Data availability: Ensures that accurate data must be available to legitimate users so that reliable access to the resources are given to the appropriate users/nodes promptly [11,57].
- Resilience to attacks: MIoT systems must avoid a single point of failure and should have the ability to adapt to node failures. In addition, there should be an underlying protection schema that protects the devices or the information in the presence of an attack [4,5].
- Data usability: Data usability ensures that only authorized users can access the data [4,5].
- Access control: There should be an underlying access control mechanism for authenticated users [17].
- Data auditing: Auditing access to medical records is an important means of controlling the utilization of resources and a standard measure for the detection and monitoring of suspicious incidents or abnormalities [5,17].
- Data authentication: This ensures the confirmation of the origin and integrity of data [9,28].
- Privacy of patient information: Medical data can be apportioned into two categories, general records and sensitive data [17]. Sensitive data can also be called patient privacy information and includes details about infectious diseases, sexual orientation, mental status, drug addiction, and identity information. Because of the criticality and the sensitivity of this data, we need to ensure that these sensitive data are not exposed to unauthorized users or that unauthorized users do not have the capacity to understand the meaning of the data, even if the data are captured and intercepted [9,28,32,45].

Readers must note that the security and privacy of patient-related data are two separate concepts [17]. Data security ensures that data are safely stored and transmitted to guarantee their confidentiality and integrity. On the other hand, data privacy implies that data can only be accessed by the people who have proper authorization to access it. Hence, the successful development and deployment of MIoT must take security and privacy both as core considerations. If not, the lack of sufficient MIoT security and privacy would not only jeopardize the privacy of patients but may also jeopardize the lives of patients [6,15,18]. In the next section, we discuss the security attacks that can be expected if these security requirements are not met, based on the MIoT layered architecture. Before further discussing the types of attack that target each layer, readers need to understand why MIoT in healthcare is becoming an appealing target for intruder attacks. Hence, in the following, based on the state of the art, we list down the key reasons why it has become an appealing target [5,6]:

- The MIoT is an emerging technological paradigm where adequate research has not been conducted regarding security, where device manufacturers themselves rush to provide MIoT solutions without security in mind.
- The fact that highly confidential and sensitive data are always being transmitted across the MIoT ecosystem makes it a sound target for attackers.
- As most of the IoT devices are inbuilt with wireless communication capabilities, it puts most MIoT devices at risk for WSN security violations [4,5].
- In order to control, monitor, and operate, MIoT solutions encompass different applications. There is huge concern about the implementation risks in this application layer, such as breaches of access control and session hijacking as well as the general security functionalities of the applications.
- A large fraction of computational resources are consumed by certain security computations such as the execution of encryption algorithms. Due to the limited computing capacities (e.g., limited computing power and memory), many of these MIoT

devices lack integrated encryption mechanisms, as execution cannot be completed in those resource-constrained environments. This lack of strong encryption mechanisms across devices makes devices susceptible to malicious attacks.

- The amount of financial profit that can be gained by exploiting the devices and data make these data a sound target for attackers by way of blackmailing someone, releasing the exploited data to the public, or selling it on the dark web [11].
- Personally identifiable information (PII) and personal health information (PHI), which can be contained within MIoT data, would make the entire MIoT ecosystem a sound target that could be exploited for profit [4].

As of now, according to the latest trends that have been witnessed, there have been various simulations and demonstrations of intruders attempting to insert malicious code directly into wearable devices using e programmable device interface or by trying to plant malware remotely to compromise the device and then obtain the sensitive data, monitor the device remotely, or control the device remotely, resulting in life-threatening circumstances [13]. Barnaby Jack demonstrated the hacking of an insulin pump at the McAfee conference in 2011 by overriding the device controls [18,50] to inject lethal insulin doses into the pump. Additionally, at the Melbourne Breakpoint security conference in 2012, he showed that a pacemaker transmitter could be reverse-engineered and hacked to produce a lethal electrical shock with a high voltage of 830 volts [18,51], resulting in a simulated cardiac arrest, clearly showing the repercussions of various vulnerabilities present with the MIoT. Moreover, according to *Wired* magazine [57–59], students at the University of Alabama demonstrated that they could hack the pacemaker in a robotic dummy patient and kill it theoretically. In 2016, a weakness in a St. Jude Medical cardiac device was discovered, where an intruder could send repetitive messages to the system until the battery was exhausted, a weakness that could eventually endanger the lives of patients [57–59]. While these examples illustrate the attacks that can be carried out on devices implanted in the human body, there are numerous vulnerabilities that are present on medical networks or terminal databases that have the ability to cause serious damage if the risk is not managed [18,56,57]. The following figure, Figure 5, depicts the top security threats that target healthcare cloud environments [57–65].

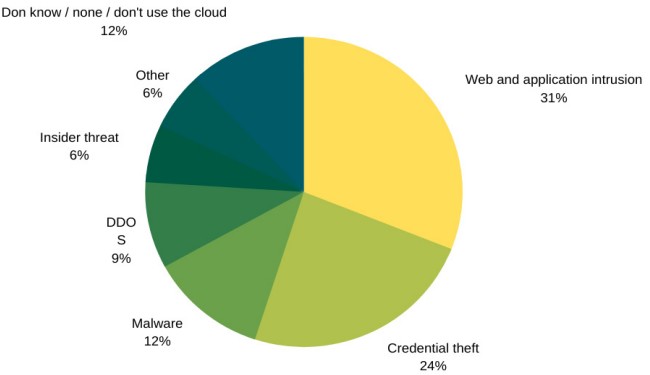

**Figure 5.** Top security threats to healthcare cloud environments.

On the other hand, recent studies indicate that at least one security breach has been reported by nearly 90% of healthcare organizations involving MIoT devices [52]. Around 45% of all ransomware attacks that occurred in 2017 targeted healthcare organizations [53]. For example, the medical records in an Indiana hospital in the USA were encrypted

by attackers, forcing the hospital to pay a USD 50,000 ransom to recover the data in 2017. By October 2016, 14 hospitals had reported ransomware attacks that had used medical devices as a gateway [58,59]. With more than 200,000 devices around the world, the biggest ransomware attack on medical systems was recorded in 2017 [12,54–56] and was known as WannaCry. It exploited vulnerabilities in the Windows OS and prevented medical staff from accessing affected computers [59–65], thus delaying critical patient care. Those ransomware attacks and hacking demonstrations imply that there is a high possibility of MIoT devices becoming compromised due to a lack of inbuilt security and a lack of user knowledge. In addition, the lack of environmental configurations in the MIoT ecosystem can also put MIoT devices at risk. Based on the latest statistics, it is evident that ransomware holds the first place among many other root causes that are responsible for the majority of healthcare data breaches, as depicted in Figure 6 [63,64,64–74].

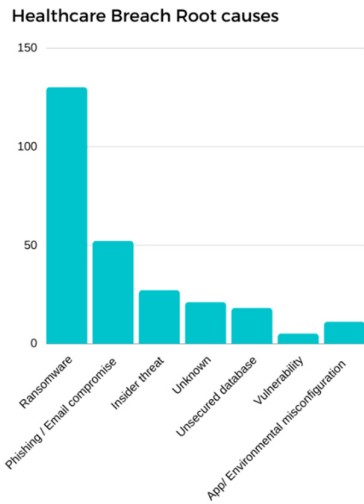

**Figure 6.** Root causes of healthcare data breaches.

## 4. Attack Classification Based on the Architecture

The intention of this subsection is to highlight the attacks targeting the MIoT ecosystem. Based on the state of the art, we have separated the attacks in terms of the MIoT layered architecture, into three sections, as in Figure 7 [18]. This is so that readers will have a better understanding of MIoT attacks and their implications as they pertain to each layer.

| Attacks on perception layer | Attacks that taget network layer | Attack that target application layer |
|---|---|---|
| • Tampering of devices<br>• Side channel attack<br>• Tag cloning<br>• Sensor tracking<br>• Insertion of forged nodes | • Denial of Service (DOS)<br>• Distributed Denial of Service (DDOS)<br>• Rogue access<br>• Eavesdropping<br>• Man in the Middle attack (MITM)<br>• Sybil Attack<br>• Sniffing Attack<br>• Routing attacks | • Session hijacking<br>• Cross-site scripting (XSS)<br>• Cross-Site request forgery (CSRF)<br>• SQL injection<br>• Brute Force attack<br>• Ransomware<br>• Buffer Overflow<br>• Phishing Attack |

**Figure 7.** Classification of security and privacy attacks in terms of the MIoT layered architecture.

### 4.1. Attacks on Perception Layer

The integrity and the privacy of data are compromised by attacks on these perception layer devices and can thus lead to adverse outcomes that can be fatal. Possible attacks that target the perception layer are listed below:

- **Tampering of devices**

  The attacker can tamper physical MIoT sensing devices and can manipulate their functionality or can fully or partially stop their functionality [4,5]. Some of the firmware vulnerabilities allow attackers to further exploit these vulnerabilities, allowing them to then implant malware on the physical MIoT device and take control of the device [57].

- **Side channel attack**

  It should be noted that attackers can utilize many techniques to perform a side-channel attack, such as monitoring the electromagnetic activity around the medical devices, by analyzing the power consumption and data movement timing [4,5,48]. A successful side-channel attack may lead to the exposure of underline confidential data.

- **Tag cloning**

  Here, the attacker can use data obtained from a successful side channel attack or can duplicate the data from a pre-existing tag [48] to perform the attack. Unauthorized data, such as confidential patient information, is able to be accessed through the help of the cloned tag after the attack [4].

- **Sensor tracking**

  During this kind of attack, attackers can exploit real-time location service devices to obtain patient location, which violates patient privacy [4]. These devices contain GPS tracking sensors to send the location of the patient in case of an emergency. If the device is vulnerable, the attacker may spoof the GPS data and will be able to determine the patient location [56,57].

- **Insertion of forged nodes**

  To gain access and to gain further control over the MIoT network, an attacker can insert a falsified or malicious node between the actual network nodes in the MIoT network [5,13].

### 4.2. Attacks on Network Layer

- **Denial of Service (DOS)**

  Medical IoT devices may have very limited capacity and capabilities, owing to the default miniaturized nature of the IoT. Hence, the attacker can use a successful DOS attack to interrupt the services performed by a MIoT network, endangering and delaying critical patient care [5]. This attack floods the system with a vast amount of service requests and disrupts the device functionality on the network [9]. Distributed denial of service (DDOS) is an aggressive form of a denial of service attack [55–57] and uses a larger number of compromised nodes to flood the system, making it more difficult to determine the original source of the attack. Attackers can use automated tools such as botnets, which are comprised of infected IoT devices, to launch a wide variety of DDOS attacks [4]. (e.g., Telnet, Mirai)

- **Rogue access**

  Here, the attacker sets up a forged gateway and lures legitimate users to connect to the rogue access point and then intercepts the network traffic [4,13] to reveal the transmitting data.

- **Eavesdropping**

  The attacker first locates and intercepts the appropriate hardware devices so that he/she is able to successfully collect the data being transmitted through hardware devices. This unlawfully obtained data can be used to conduct different forms of attacks. Although this problem can be solved by encryption, strong encryption, particularly with low-powered MIoT devices, is not always practical due to a lack of processing power and memory [4,5,74–79].

- **Man in the Middle attack (MITM)**

  The MITM attack allows the attacker to exploit a possible vulnerability and view and listen to the data and thereafter secretly replay and alter the data that is being communicated. Since data are sent and retrieved by MIoT sensing devices, any modifications made to the data during transmission can lead to mistreatment (e.g., medication overdose) [5,13].

- **Replay attack**

  In this type of attack, an attacker is able to reuse a message that was previously shared between legitimate users for authentication. By breaching any of the network nodes or by eavesdropping, it is possible for the intruder to intercept the authentication message [5,13,57].

- **Sybil attack**

  This is a common attack that targets WSNs. A node in the network system provides the victim node with multiple identities, allowing the victim node to perform a single operation multiple times. As the attacker has multiple identities in the WSN, the victim node will transmit data through the compromised nodes exposing, the sensitive data [13,67].

- **Sniffing attack**

  Using sniffing devices or applications, the attacker tries to thieve or intercept the data in the network traffic and collect useful information for further attacks [4,5,66,67].

- **Routing attacks**

  The way that messages or data are routed is affected by this form of attack. The attacker may redirect, misdirect, spoof, or even drop the packets at the network layer in this form of attack [14].

### 4.3. Attacks on Application Layer

In the application layer, attacks primarily seek unauthorized access to sensitive user data, which ultimately violates user privacy. Attackers usually take advantage of software and device bugs (e.g., buffer overflow, code injection) on the application layer to compromise the services and applications offered by the application layer. In addition to these attacks, different forms of malware such as worms, viruses, and trojans often threaten applications and services. Further, other malicious programs (adware, key loggers, rootkit, and spyware, etc.) often undermine the privacy of the users [13]. In the following list, we discuss potential application-layer attacks:

- **Session hijacking**

  Session hijacking is subjected to critical vulnerabilities in the session connection at the application interface, where an intruder can hijack the program session and can gain control over the application controls [4,5,13].

- **Cross-Site Scripting (XSS)**

  Cross-site scripting attacks exploit applications by inserting malicious scripts to bypass access control through web pages, (e.g., web control pages) [13].

- **Cross-Site Request Forgery (CSRF)**

  In CSRF attacks, the attacker forces an end user to execute unwanted actions on a web application on which they are currently authenticated, leading to devastating results such as revealing user credentials [4,13].

- **SQL injection**

  In SQL injection attacks, the attacker attempts to attack the application-connected backend database by inserting malicious SQL statements. A successful SQL attack will lead the attacker to the backend database, where the attacker can exploit all of the critical patient data stored in the database [13].

- **Brute force attack**

  Because of the weaker computational capacity possessed by most of the MIoT devices in a medical network, a simple brute force attack can easily compromise the device's access control and can open ways for attackers to further compromise the network, such as through planting malware on the devices [4,5].

- **Ransomware**

  Ransomware encrypts all of the data in a system and asks for a ransom to be paid in order to redeem the compromised system. If appropriate security settings are not placed, ransomware may also start with one single compromised victim machine and may then spread across the entire network [4,5,18,51,79–85].

- **Buffer Overflow**

  A buffer acts as a temporary area for data storage. When a software or device operation puts more data into the buffer, the extra data may overflow. This allows some of the information to leak into other buffers that may corrupt or overwrite any of the information that they carry. In a buffer overflow attack, the extra data sometimes hold specific instructions for actions intended by a hacker or malicious user (e.g., the data may trigger a reaction that destroys files, granting admin privileges, alters data, or exposes private information) [13,68].

- **Phishing attack**

  An attacker pretends to be a legitimate person or an entity in a standard phishing attack and tries to access personal information, such as credit card data and user credentials. Email is extremely common as a medium that circulates these kinds of phishing attacks, where the attacker obtains confidential information when the user opens the email or email attachment [4,5,13].

Based on the literature that we have reviewed, it should be noted that there are a wide range of attacks, and they can take place on any of the layers. As such, we need to secure the entire MIoT ecosystem, not just specific technologies pertaining to one single layer, if we want to ensure optimal security [13,86–98].

## 5. Related Work and Contributions

In this section, we mainly highlight what exact contributions have been made by other researchers towards the state of the art by highlighting the title, scope of the study, key findings along with our observations, and whether the study is focusing on the IoT in general or specifically on MIoT. What we have understood is that even though there are previous studies related to MIoT security and privacy, none of the studies have been able to highlight security attacks in terms of the layered architecture of the MIoT, and none of them were able to highlight countermeasures and solutions in terms of recent security and privacy issues and that are related to the layered architecture in general. Hence, we believe that this study will be highly beneficial for researchers who are keen on learning about this subject. In the following table, Table 6, we highlight the contributions made by others towards the state of the art, which can be used to compare the current level of the status of the state of the art.

**Table 6.** Summary of contributions.

| Reference and Year | Tittle | IoT in General | MIoT Specific | Scope | Contributions and Critique |
|---|---|---|---|---|---|
| [4], 2020 | The Role of the Internet of Things in Health Care: A Systematic and Comprehensive Study | | ✓ | A general systematic review that highlights security and privacy challenges. | The researchers discussed recent security and privacy challenges, pointing out the vulnerabilities that are present, but they did not put much effort towards discussing security and privacy requirements, countermeasures, and solutions. |
| [6], 2017 | Security and privacy in the internet of medical things: taxonomy and risk assessment | | ✓ | Taxonomy of security and privacy issues pertaining to MIoT is discussed. | A quantitative approach to identifying and assessing risks in MIoT is highlighted. Even though the researchers highlighted security and privacy issues, how to mitigate them was not discussed in detail in the study. |
| [7], 2017 | Towards composable threat assessment for medical IoT (MIoT) | | ✓ | A general analysis of MIoT threat assessment is discussed. | A framework for identifying, assessing, and evaluating threats in the MIoT environment is highlighted, whereas the study did not provide adequate knowledge about the security and privacy of MIoT. |
| [8], 2018 | Secure medical data transmission model for IoT-based healthcare systems | | ✓ | Proposed a hybrid security model for securing the content in medical images. | A security model based on the steganography technique with a hybrid encryption scheme is introduced towards protecting the security and data integrity of patient diagnosis data that are transmitted across MIoT networks. |

| [9], 2012 | Internet of Things in healthcare: Interoperatibility and security issues | ✓ | A general discussion with regard to the security issues, benefits, and solutions in MIoT is provided. | Security challenges pertaining to the IoT in telemonitoring are highlighted. Even though the researchers provided a discussion in terms of the security of telemonitoring, this study did not provide adequate knowledge related to security attacks pertaining to MIoT-based telemonitoring solutions. |
|---|---|---|---|---|
| [10], 2018 | An internet of things-based health prescription assistant and its security system design | ✓ | A theoretical framework for an MIoT health prescription assistant is proposed. | A security system for a health prescription assistance system is designed, implemented, and validated; this study only provides knowledge about the security aspects of MIoT-based telemedicine and not about the entire MIoT environment. |
| [11], 2019 | A joint resource-aware and medical data security framework for wearable healthcare systems | ✓ | A security framework for resource-constrained wearable health monitoring systems is introduced. | A biometric-based security framework for a wearable health monitoring systems is introduced, and a performance comparison of the proposed model is also conducted. Even though the researchers conducted an experimental evaluation to test their framework, they failed to discuss security and privacy repercussions in terms of wearable MIoT, which is becoming a booming trend as of now. |
| [12], 2019 | IoMT-SAF: Internet of medical things security assessment framework | ✓ | A web-based MIoT security assessment framework is developed. | The researchers developed a web-based framework that recommends security features in MIoT and assesses protection and deterrence based on ontology. Nevertheless, even though the featured web solution ranks the security solutions in terms of security and privacy, the study does not provide comprehensive knowledge about MIoT security and privacy. |
| [13], 2018 | Internet-of-Things security and vulnerabilities: Taxonomy, challenges, and practice | ✓ | A summary of overall IoT security attacks is depicted. | The authors provide a taxonomy and classification based on IoT security attacks based on application domains, including healthcare, where they also highlight security and privacy requirements. Even though this provides a comprehensive overview of security and privacy attacks, countermeasures for the attacks are not featured in this study. |
| [17], 2018 | Security and privacy in the medical internet of things: a review | ✓ | A review on security and privacy requirements and | Security and privacy requirements, open challenges, and anticipated future directions in terms of the security and privacy of MIoT are discussed, where this provides adequate knowledge |

| | | | | |
|---|---|---|---|---|
| | | | solutions with regard to the MIoT is provided. | about the underlying ecosystem, leading to security vulnerabilities. |
| [18], 2015 | A review of security protocols in mHealth wireless body area networks (WBAN) | ✓ | Threats pertaining to the medical networks are discussed. | The latest trends and future directions are discussed as they pertain to medical networks, where the researchers were able to provide adequate knowledge about security requirements and mechanisms pertaining to medical networks. However, this features the security of MIoT networks only. |
| [21], 2016 | A secure IoT-based healthcare system with body sensor networks | ✓ | A secure MIoT system is proposed in this study. | A secure MIoT system with a body sensor network is proposed and implemented using the Raspberry PI platform. |
| [28], 2015 | BSN-Care: A secure IoT-based modern healthcare system using body sensor network | ✓ | Security concerns and requirements in the field of body sensor network-based healthcare systems are discussed. | The researchers proposed a secure MIoT healthcare system using a body sensor network (BSN) called BSN-Care. |
| [35], 2017 | Internet of Medical Things (IOMT): applications, benefits and future challenges in healthcare domain | ✓ | Applications and challenges of MIoT are discussed, highlighting the security concerns. | A comprehensive review is presented in terms of MIoT applications and challenges to the domain, where the researchers do not put much emphasis on security as a key challenge. |
| [36], 2017 | Internet of things for smart healthcare: Technologies, challenges, and opportunities | ✓ | A survey is conducted that highlights the state of the art related to the MIoT. | Security, privacy, wearability, and low-power operations related to MIoT are discussed, but the researchers do not provide that much focus on security and piracy as a key challenge. |
| [41], 2016 | Security context framework for distributed healthcare IoT platform | ✓ | Context aware security framework is introduced for MIoT | The researchers introduce a security framework that can be used to secure data transmission across distributed MIoT platforms, whereas this does not feature any security and privacy attacks. |
| [43], 2014 | The internet of things for healthcare monitoring: security review and proposed solution | ✓ | Security problems related to MIoT monitoring system are presented. | Proposed a security model for MIoT monitoring systems based on symmetric cryptography and network node authentication mechanisms, and authors provide decent knowledge in terms of security arracks and countermeasures. |
| [44], 2016 | The internet of things in healthcare: Potential applications and challenges | ✓ | A review is provided in terms of MIoT applications. | Challenges are discussed in terms of the security and privacy MIoT applications as they pertain to MIoT solutions. |

| | | | |
|---|---|---|---|
| [47], 2013 | Wattsupdoc: Power side channels to nonintrusively discover untargeted malware on embedded medical devices | ✓ | A research experiment is conducted with regard to MIoT platforms. | Proposed an add-on monitoring system for detecting malware that targets MIoT devices. |
| [60], 2015 | The internet of things for health care: a comprehensive survey | ✓ | A survey is provided in terms of architecture, threat model, security requirements, and challenges. | Security and privacy issues, the latest trends pertaining to the MIoT ecosystem, are highlighted; this study provides a comprehensive survey about pervasive MIoT ecosystems. On the other hand, this study does not place much focus on security and privacy as key challenges. |
| [63], 2019 | Security Requirements of Internet of Things-Based Healthcare System: a Survey | ✓ | In order to identify the security requirements pertaining to MIoT, a survey is conducted. | Features and concepts associated with security requirements for MIoT are highlighted, but this study only features security and privacy requirements. |
| [70], 2018 | Security Threats and Recommendation in IoT Healthcare | ✓ | Provides a review of various MIoT systems. | Security and privacy issues pertaining to MIoT systems and threats are discussed in terms of the layered architecture, and the researchers do not place much emphasis on challenges and future directions in terms of security and privacy. |
| [85], 2020 | A secure fuzzy extractor based biometric key authentication scheme for body sensor network in Internet of Medical Things | ✓ | Biometric authentication schema for MIoT body sensor network is introduced | Secure fuzzy extractor combined with fuzzy vault is introduced to protect sensitive patient data using encryption techniques. |
| [91], 2021 | A privacy and session key based authentication scheme for medical IoT networks | ✓ | Network security schema for MIoT is introduced. | Secure addressing and mutual authentication protocol (SAMA) scheme is proposed and validated using formal and informal methods. |
| [92], 202. | Security and Privacy in IoT Smart Healthcare | ✓ | The current state of security and privacy of MIoT is analyzed. | The researchers provide a detailed discussion about challenges and security frameworks in terms of MIoT, and they also highlight security solutions in the study. |
| [93], 2021 | Towards design and implementation of security and privacy framework for internet of medical things (iomt) by leveraging blockchain and ipfs technology | ✓ | Smart-contract-enabled blockchain network is introduced | To protect the security and privacy of patient data, a blockchain-based methodology is proposed where the data can be stored on blockchain ledgers, towards enhancing data integrity and user privacy. |

Based on the review that we conducted, when compiling this summary of contributions, we noted that most of the reviews and surveys only focused on reviewing the attack types and requirements and not solution. As such, the number of papers on the subject of solving the problems related to different attack types are minimal. On the other hand,

most of these papers lack deep analysis about the security and privacy issues pertaining to the MIoT layered architecture. Hence, in order to fill this gap, in this study, we discuss the prevailing countermeasures and solutions in terms of layered architecture and also highlight challenges and future directions as well. Table 7 provides a comparison of the existing research work within our study for better understanding. In a nutshell, our study fulfills all of the criteria, which defines in the comparison table, which signifies our work among others.

**Table 7.** A comparison of related work with our study.

| Reference and Year | Security and Privacy Requirements Are Discussed | Security and Privacy Attack Types Are Discussed | Countermeasures and Solutions for Attacks Are Discussed | Challenges Are Discussed in Terms of Security and Privacy | Recent Trends in Terms of Security and Privacy Are Highlighted | Future Directions Are Discussed in Terms of Security and Privacy |
|---|---|---|---|---|---|---|
| Thilakarathne, N. N., Kagita, M. K., and Gadekallu, D. T. R. (2020) | X | X | X | ✓ | ✓ | X |
| Alsubaei, F., Abuhussein, A., and Shiva, S. (2017) | ✓ | X | X | X | X | X |
| Darwish, S., Nouretdinov, I., and Wolthusen, S. D. (2017) | ✓ | X | X | ✓ | X | X |
| Tarouco et al. (2012) | ✓ | X | X | ✓ | X | X |
| Chenet al. (2018) | ✓ | ✓ | X | X | X | X |
| Sun et al. (2018) | ✓ | ✓ | ✓ | X | X | ✓ |
| Joyia et al. (2017) | X | X | X | ✓ | X | X |
| Baker, S. B., Xiang, W., and Atkinson, I. (2017). | X | ✓ | X | ✓ | X | X |
| Rghioui et al. (2014) | ✓ | ✓ | ✓ | X | X | X |
| Laplante, P. A., and Laplante, N. (2016) | X | ✓ | ✓ | ✓ | X | X |
| Islam et al. (2015) | X | ✓ | ✓ | ✓ | X | X |
| Nasiri S, Sadoughi F, Tadayon MH, and Dehnad A (2019) | ✓ | X | X | ✓ | X | X |
| Karunarathne et al. (2021). | ✓ | X | X | ✓ | X | X |
| Our review | ✓ | ✓ | ✓ | ✓ | ✓ | ✓ |

## 6. Countermeasures and Solutions

The Open Web Application Security Project, or most popularly known as OWASP [69], has released a secure medical device deployment guide that was developed in conjunction with the Cloud Security Alliance. It provides a comprehensive overview about what kind of controls and precautions should be taken in order to strengthen the security of a medical IoT environment [4,62–70], such as having perimeter defense mechanisms, network security controls, device and OS update guidelines, device security controls, security testing plans (e.g., penetration testing), and proper incident response plans. In addition, the OWASP Internet of Things project [4,5] provides specific guidance for manufacturers, developers, and consumers to better understand the security challenges associated with the IoT and to allow them to make better security decisions when designing, deploying, or evaluating IoT technologies in any context [4,69,70].In order to guarantee better security and privacy within the entire MIoT ecosystem, the following criteria must be satisfied, and they should be integrated during the development and the deployment process of MIoT components that can be used to mitigate most of the threats [64,70–74].

- **Access control**

  Access control specifies who has been authorized to access the medical data and MIoT devices, and how much access they are allowed and should be granted. Hence, it should verify the identity of the party attempting to access the data (e.g., using a password, fingerprint, etc.) [17,36]. As for that, well-designed access control must be implemented for IoT healthcare applications as well as devices to ensure maximum security and privacy [10,21,71]. On the other hand, when it comes to physical security, which is another aspect of access control, it should consider MIoT devices and medical data against physical theft, accidents, environmental hazards, and sabotage whenever it is necessary, adhering to all security and privacy requirements [70].

- **Data encryption**

  Data encryption during data storage or transmission provides protection for the data. Solid data encryption would make it difficult for an attacker to read sensitive health data even if the attacker has access to the medical database or transmission media [20,36,41,64,71].

- **Data auditing**

  Audits are very useful to determine the source of any security breach [19], allowing the underlying information to be examined (e.g., system notifications, network traffic, user access, etc.). On the other hand, the medical IoT network is extended towards the cloud and cannot be fully trusted. Hence, it is highly required to have an auditing mechanism in the cloud in order to identify the disruptions and anomalies happening across the cloud network [17].

- **IoT healthcare policies**

  Policies and regulations play a pivotal role when transforming the healthcare sector to the next level by imposing various standards and regulations that everyone must comply with. As such, the United States has the Health Insurance Portability and Accountability Act (HIPAA), which was introduced in 1996, which defines the standards for protecting sensitive patient data and measures to be followed by any organization that deals with protected health information [19], whereas systems in the European countries must comply with Data Protection Directive, which was introduced in 1995.

- **Data search**

  In order to preserve data privacy, confidential data must be encrypted prior to outsourcing, which outdates conventional data usage based on plaintext keyword searches [5,17].

- **Data minimization**

  Data minimization suggests that the services provided by the Medical IoT should limit the collection of personal health information (PHI) to only the information that is needed, and they should also only retain the data for as long as is necessary to fulfill the purpose of the services that the users are requesting. Effective data minimization techniques in healthcare include minimizing the overall amount of patient personal data that are collected. On the other hand, only collecting the adequate and relevant amount of patient data and the amount that is in line with the intended purpose; deletion or masking obsolete or unnecessary personal data that are no longer needed; conducting periodic checkups to ensure the adequacy and relevance of the data that are collected are the other techniques that can be employed. As too much personal data may bring bigger risks, the effective utilization of data minimization would also help to lower the risks as well as lower the storage cost [5,21,25,45].

- **Data anonymization**

  Data anonymization means the process of protecting private or sensitive information by erasing or encrypting identifiers that connect an individual to stored data. As an example, one can run a personally identifiable information (PII) information such as their name, social security number, and address through a data anonymization process that retains the data but keeps the source of the data anonymous [5,8,34].

- **Inventory devices**

  Since healthcare organizations cannot secure what they cannot see, it is essential to create a full map of all the organizational assets. Many IoT devices are brought in without a proper risk assessment; hence, regular risk assessment must be conducted in order to identify potential risks. Some vendors provide inventory tools that can identify IoT devices on the network without disrupting their functionality [64,71].

- **Network segmentation**

  System administrators in the organization must segment public networks from the rest of networks, limit access to virtual LAN assets and events, or segregate department-based traffic for providing effective organization wide security [64,71].

- **Follow the best practices**

  Maintaining the best security practices, such as avoiding hard-coded passwords, deploying firewalls, and honeypots for luring and mitigating the attackers, and encrypting confidential data are highly essential when it comes to improving security in a typical MIoT setting. On the other hand, as these MIoT devices and applications are continuously being connected to networks, implementing intrusion prevention systems (IPSs), intrusion detection systems (IDSs), security sockets layer/transport layer security (SSL/TSL), and hypertext transfer protocol secure (HTTPS) communication mechanisms should be used in order to ensure network security [70]. Furthermore, before deploying devices, a risk assessment has to be completed in order to understand what vulnerabilities exist before setting up the environment. Moreover, devices and software must be updated regularly [64,71].

- **Wider awareness**

  In general, employers in healthcare should have the necessary awareness of the principles of information security in order to provide security and necessary protection for MIoT applications and confidential patient data, and providing continuous medical staff training is essential towards improving the safety and wellbeing of patients. Staff training should comprise of providing them with adequate knowledge in data security and privacy and patient rights [70].

- **Continuous monitoring and reporting**

  All MIoT applications and device-related logs must be collected to a centralized log management system for continuous network monitoring and Internet attacks. By

having a central log management system, logs can be monitored, analyzed, and evaluated in real-time with the help of artificial intelligence (AI)-powered machine learning and deep learning techniques, resulting in preventing any security incidents from happening. This could be completed by implementing an organization-wide security information and event management system (SIEM), which would help to prevent attacks prior to when they are onset and give the capacity to respond to security incidents in a strong way in real time [3,4,70].

Next, based on the MIoT attacks that we have discussed, Table 8 provides a comprehensive summary of what sorts of countermeasures can be taken towards preventing attacks in terms of the MIoT layered architecture along with other types of measures that can be taken, along with our justification.

**Table 8.** Summary of countermeasures and solutions that can take against MIoT attacks, in terms of its layered architecture.

| MIoT Attack Type | Related Layer | Access Control | Data Encryption | Data Auditing | IoT Healthcare Policies | Data Search | Data Minimization | Data Anonymization | Inventory Devices | Network Segmentation | Follow the Best Practices | Wider Awareness | Continuous Monitoring and Reporting | Justification/Other Measures That Can Take |
|---|---|---|---|---|---|---|---|---|---|---|---|---|---|---|
| Tampering of devices | Perception | ✓ | | | ✓ | | | | ✓ | ✓ | ✓ | | ✓ | As the tampering of devices deals with physical MIoT components, having access control mechanisms, anti-tampering mechanisms, implementing organization-wide healthcare policies and regularly monitoring the devices, and following up the best security practices can be taken as countermeasures. |
| Side channel attack | Perception | | ✓ | | ✓ | | ✓ | ✓ | | ✓ | ✓ | | ✓ | As side-channel attacks are reliant on the relationship between information leaked through a side-channel and secret data, two types of countermeasures can be taken: eliminating or reducing the release of such information and eliminating the relationship between the leaked information and the secret data using some kind of data scrambling method. |
| Tag cloning | Perception | ✓ | ✓ | | ✓ | ✓ | ✓ | ✓ | | ✓ | ✓ | | ✓ | Following a successful side-channel attack, a tag cloning attack can be performed, |

| Attack | Layer | | | | | | | | | | | | Countermeasures |
|---|---|---|---|---|---|---|---|---|---|---|---|---|---|
| | | | | | | | | | | | | | which can be mainly mitigated through the implementation of data encryption methods. |
| Sensor tracking | Perception | ✓ | ✓ | | ✓ | ✓ | ✓ | ✓ | ✓ | | ✓ | ✓ | ✓ | In order to avoid sensor tracking attacks, data that are mainly transmitted across MIoT networks can be encrypted, and access control mechanisms can be implemented. On the other hand, data search, anonymization, and minimization techniques can also be used. |
| Insertion of forged nodes | Perception | ✓ | | | ✓ | | | ✓ | | | ✓ | ✓ | ✓ | In order to avoid these types of attacks physical access control mechanisms, regularly monitoring devices can be conducted apart from adhering to healthcare policies and by following up with the best practices. |
| Denial of Service (DOS) | Network | | | | ✓ | | | | ✓ | ✓ | ✓ | | ✓ | In order to prevent DOS attacks, network security can be strengthened by implementing next-generation firewalls, IPS, and IDS systems. Nevertheless, routers and firewalls can be configured to block malicious traffic, and unnecessary TCP/UDP services can be blocked to prevent DOS attacks. |
| Distributed Denial of Service (DDOS) | Network | | | | ✓ | | | | ✓ | ✓ | ✓ | | ✓ | The same measures that are used to counter DOS attacks can be used as countermeasures for DDOS attacks. |
| Rogue access | Network | | ✓ | | ✓ | | ✓ | ✓ | | ✓ | ✓ | | ✓ | Apart from following up the best security practices and increasing user awareness, wireless intrusion prevention systems can be implemented to monitor the radio spectrum for unauthorized access points. |
| Eavesdropping | Network | | ✓ | | ✓ | | ✓ | ✓ | | ✓ | ✓ | | ✓ | To prevent eavesdropping attacks, medical data can be encrypted. On the other hand, by using a personal firewall, keeping antivirus |

| | | | | | | | | | | |
|---|---|---|---|---|---|---|---|---|---|---|
| | | | | | | | | | | software updated, and using a virtual private network, these attacks can be prevented. |
| Man in the Middle attack (MITM) | Network | | ✓ | ✓ | ✓ | ✓ | ✓ | ✓ | ✓ | Encrypting the medical data that transmit across the network and sticking with the best security practices would help to prevent MITM attacks. |
| Sybil Attack | Network | ✓ | ✓ | ✓ | | | ✓ | ✓ | ✓ | These attacks can be prevented by implementing ace control mechanisms and by following up with the best security practices. |
| Sniffing Attack | Network | | ✓ | ✓ | ✓ | ✓ | ✓ | ✓ | ✓ | These attacks can be prevented using implementing access control mechanisms and by following up the best security practices. |
| Routing attacks | Network | | ✓ | ✓ | ✓ | ✓ | ✓ | ✓ | ✓ | Routing attacks can be prevented using implementing network security mechanisms such as IDS. |
| Session hijacking | Application | | | | | | ✓ | ✓ | ✓ | In order to prevent session hijacking attacks, the network data can be encrypted, and other network security protection mechanisms such as SSL/TLS and HTTPS schema can be implemented to secure the communication media. |
| Cross-site scripting (XSS) | Application | ✓ | | | | | ✓ | ✓ | ✓ | XSS attacks can be prevented by having adequate application access control mechanisms, validating user inputs, and encoding output data. |
| Cross-Site request forgery (CSRF) | Application | ✓ | | ✓ | | | ✓ | ✓ | ✓ | CSRF attacks can be prevented by having adequate network security mechanisms and by using anti-CSRF mechanisms. |
| SQL injection | Application | ✓ | | ✓ | | | ✓ | ✓ | ✓ | These attacks can be prevented by implementing web application firewalls (WAF), IDS, and by having adequate application security controls, such as input validations. |

| Attack | Layer | | | | | | | | | Prevention |
|---|---|---|---|---|---|---|---|---|---|---|
| Brute Force attack | Application | ✓ | ✓ | | ✓ | ✓ | | ✓ | ✓ | ✓ | Brute force attacks can be prevented by sticking with healthcare security policies, using adequate application security controls, and continuously monitoring logs through a SIEM. |
| Ransomware | Application | | | ✓ | | | | ✓ | ✓ | ✓ | These attacks can be mitigated by increasing user awareness and by sticking with the best security practices, such as having anti-virus and anti-spyware solutions and using the inbuilt ransomware protection features of operating systems. |
| Buffer Overflow | Application | ✓ | | | | | | ✓ | ✓ | ✓ | These attacks can be prevented by disabling unnecessary network services and by implementing firewalls and IPS systems. |
| Phishing Attack | Application | | ✓ | ✓ | ✓ | ✓ | | ✓ | ✓ | ✓ | In order to prevent phishing attacks, user awareness can be improved as the first defense, and antivirus and antispyware solutions can be installed on end-user machines to protect the organization's resources. |

## 7. Challenges and Future Directions

The growing malware attacks pertaining the range of MIoT devices in healthcare are expected to rise in the forthcoming years, resulting in higher demand for robust IoT security. Among these malware attacks, ransomware and DDOS attacks are abundant. On the other hand, the latest studies and research indicate that the healthcare industry has a natural appeal to ransomware attacks due to the poor security configurations placed in the entire ecosystem. Nevertheless, placing appropriate and adequate security controls are challenging due to various factors. Hence, several challenges pertaining to the MIoT ecosystem need special attention in order to create stronger security within the MIoT ecosystem, where they would hinder the way of devising perfect security solutions. In the follow list, we discuss the main challenges that pertain to MIoT security and privacy and that need to be addressed with urgency [4,5,64,70–94].

- **Insecure network**

    Due to convenience, high availability, and low cost, most medical IoT devices are rely heavily on wireless networks, such as WI-FI, which pose major security vulnerabilities, such as a factory fitted default username and passwords and weak authentication methods, which are a potential target for network-level attacks, such as sniffing, eavesdropping, and WI-FI password cracking attacks [3–5,70,71].

- **Limitation of resources**

    In general, IoT healthcare devices are vary depending on their manufacturer, size, and complexity. When it comes to the internal design, most of the devices may have

low-speed processors, low inbuilt memory, and storage capacities. Due to the constrained nature of the resources of most medical IoT devices, even a simple brute force attack can easily exploit and compromise the access control of such devices, leading to a mega comprise in the entire MIoT healthcare network. Hence, due to this resource-constrained nature and the complexity of device manufacturers, devising security solutions that minimize resource consumption over execution and that maximize security efficiency is a huge challenge [64,70–79].

- **Heterogeneous devices**

  When it comes to most medical IoT devices, even devices that are made for a specific purpose will change based on the device manufacturer, as there is no constant or agreed-upon standards between device manufactures. Therefore, a MIoT device made for a specific purpose by manufacturer A would not be matched with a device made by manufacturer B, which increases the complexity of the devices, thus posing a challenge regarding devising unified secure schemes pertaining to the entire MIoT ecosystem [79–84].

- **Zero-day vulnerabilities and security patches**

  Due to the intrinsic ubiquitous nature and rapidly changing threats, MIoT devices are highly likely to be exploited by zero-day vulnerabilities, which creates doubts about regularly updating devices to patch potential vulnerabilities before malicious attackers try to exploit them. Intruders, on the other hand, are always looking for weak places or weak links to exploit. For example, the outdated programs that are commonly found in the application layer are the most exposed to security attacks. Similarly, healthcare system providers seldom deliver the most recent firmware upgrades to physical MIoT devices, leaving end-user devices vulnerable to attack. As a result, to maintain high availability and prevent zero-day assaults, healthcare service providers should deliver regular updates to MIoT devices and applications [70,79–84].

- **High Mobility**

  In general, most MIoT devices are highly mobile in nature. For example, if a patient is wearing a wearable heart rate monitor that is connected to the internet, then the device will send data to the cloud or to the patient's caregivers based on where the person is. When the person is at the office, it will connect to the office network, and it will connect to the home network when the person is at home. Hence, devising a sound security solution that focuses on the high mobile nature of MIoT is a tedious task, as depending on the environmental security configurations, threat mitigation approaches change [64,70–74].

- **Dynamic network topology**

  Most of the MIoT devices connected to the main IoT network can leave the network either graciously (with proper notice of the exit) or disgracefully (suddenly), posing a doubt about applying universal security solutions for such complex dynamic network topologies [3–5,84–89].

- **Trust management**

  Trust management is vital element in IoT and provides necessary security and privacy to the underlying data. As all of the devices in a typical IoT healthcare network connect to the Internet to send and retrieve data, IoT devices connected to the Internet must be trusted [3,4]. On the other hand, data collection trust is becoming a serious issue due to the large volume of data collected by these MIoT devices [70]. This vast volume of data is often known as big data, and the trust affiliated with big data is becoming a huge concern in healthcare as of now. Thus, researchers are currently studying the challenges of this trust management to prevent security and privacy attacks. These trust management issues impede the functionality of the network and application layer [64,70].

- **Social Engineering**

  End-users, who are patients in this case, tend to disclose personal information openly on social media sites such as Facebook, Instagram, and others, as a result of the huge influence of social media. Cybercriminals on the other hand, perceive these sites as a new and profitable platform to distribute malware because of their enormous user base. As a result, end-users should avoid sharing personal information with strangers on these websites or over the phone [70,89–94].

Next, in Table 9, we provide a summary of challenges based on the MIoT layered architecture to provide a better understanding for our readers.

**Table 9.** Summary of challenges based on the layered architecture.

| Challenge | Impede to the Functionality of | | | Related Work |
|---|---|---|---|---|
| | Perception Layer | Network Layer | Application Layer | |
| Insecure network | | ✓ | ✓ | Research studies [2,3,6–8,10,12,18,21–23,28,32,35,41,48] mostly focus on this insecure network challenges in the MIoT environment. In this regard, the authors contributed in the form of surveys/reviews and proposed solutions for network data transmission. |
| Limitation of resources | ✓ | ✓ | ✓ | The research studies [4–7], [9,32,35,47] mostly focus on this challenge, and most of the studies were surveys/reviews. |
| Heterogeneous devices | ✓ | | | These research studies [4–7,9,23,32,35] mostly focus on this challenging aspect, and most of them were surveys/reviews. |
| Zero-day vulnerabilities and security patches | | ✓ | ✓ | These research studies [4–7,9,32] [35] mostly focus on this challenging aspect, and most of them were surveys/reviews, and some of the studies proposed novel solutions. |
| High Mobility | | ✓ | | These research studies [4–6] [9,32,35] mostly focus on this challenging aspect, and most of them were surveys/reviews. |
| Dynamic network topology | | ✓ | | These research studies [4–7,9,32,35] mostly focus on this challenging aspect, and some of the studies have proposed solutions to this problem. |

| | | | |
|---|---|---|---|
| Trust management | ✓ | ✓ | These research studies [4–7,9] [27,32,35,47,62] most focus on this challenging aspect, and some of the authors have proposed trust management solutions by incorporating encryption mechanism among these studies. |
| Social Engineering | | ✓ | Even though this was a rarely spoken subject, the research studies [4–7,9,32,35] provide some clues about this challenging aspect. |

By highlighting the main challenges that hinder the ways of devising sound security solutions, next, we will discuss the anticipated future directions of MIoT that we can see in coming years.

*Future Directions*

In a nutshell, medical IoT-based innovations offer many useful services and applications to improve the efficiency of medical care and also facilitate improving the efficiency of healthcare facilities by delivering efficient services on-time [62]. In the following list, we highlight some of the key features and trends that we can anticipate in upcoming years in terms of Medical IoT security [14,17,28,61,63,77,94–99].

- In recent times, the emergence of AI has made a huge turning point in the IoT healthcare market and is helping to the growth of the MIoT market. Hence, it is evident that the significant use of AI-powered solutions will assist in real-time security monitoring [77,85]. Nevertheless, it is noted that MIoT security solutions rely on 03 aspects to successfully mitigate risks, such as the discovery of risks, network monitoring, and incident management, where AI would be an integral part of these solutions to provide an in-depth overview of threats and incidents and to provide the ability to respond to attacks in a timely manner while performing real-time monitoring.
- Since most medical IoT devices do not have enough computational power and memory on the devices themselves, for further data processing and storage, powerful and highly scalable computing and large storage infrastructure are needed [17,86–89]. As a result of that, due to high scalability and rapid elasticity, many healthcare organizations prefer to store their data and deploy their application servers in a stable cloud environment. As such, focus will be moved towards securing cloud environments where processing and storage happen at the same time in the same place [5,94,95].
- IoT Edge computing, which is known as data analysis, that is closer to the data source will combine with highly secured cloud environments for data storage and data processing [63,95].
- Blockchain-distributed ledger technology will be integrated with IoT to secure medical data, where the medical data and all the network transactions related data can be saved on blockchain ledgers in order to protect user privacy and data integrity [61,87,90].
- When designing MIoT devices and components, the focus will be moved towards embedded security rather and end-to-end security [88,90].
- Secure design will be an essential part of the medical IoT project development process [86,87].

- Security standards and regulations will be tighter to make sure that there are fewer gaps in the healthcare organizational security [96].
- The healthcare organizations will invest more and more in improving the organization's information security strategy in order to maintain their organizational image and to prevent any adverse security incidents beforehand [96].
- Traditional network routers provide some sorts of security capabilities such as firewall, password protection, and blocking unnecessary services, whereas modern network routers will continue to become more secure and smarter, which is a better way to provide added network security at the network entry points [96–99].
- Biometric identity management is highly valued in healthcare settings, and its use is increasing. Along with multi-factor authentication, this is presently being investigated by more advanced healthcare organizations and pharmaceutical firms in order to improve their organizational security. Biometric applications in healthcare organizations is expected to expand in the next three to five years, and this would mostly be used as a method to add an extra security layer, govern identity management and access, and provide a more seamless clinical experience to all stakeholders [96–99].

## 8. Conclusions

The literature reviewed in this study leaves no question that security should be an integral part of potential MIoT system development and deployment. The new threat environment and the criticality of the MIoT ecosystem are seen in our review. Even though many studies have suggested enhancement and novel integrations for improving MIoT security, they still do not resolve the fundamental problem, which is the lack of adequate security within systems, which creates a gap in security and privacy. The COVID-19 pandemic has showcased how healthcare can move at fast pace in order to embrace new technologies. This agility in the healthcare sector will be one of the key legacies of the pandemic. It will encourage a continuous focus on innovation as the industry looks for new ways to improve outcomes for patients and healthcare professionals across the board. What the authors have understood is that effective security needs to be built-in, not patched. It has to be an integral part of the pervasive MIoT ecosystem. Even though great attention has been paid towards MIoT security, relevant standards and technical specifications are still improving and are far from reaching the optimal maturity level. Owing to the recent demands for the security and privacy of the MIoT, many researchers are currently working towards novel secure MIoT solutions that will offer many useful services and applications to improve the efficiency of medical care while maintaining security and privacy under any context. This would be fueled by the integration of AI, blockchain, essential secure design, focus on embedded security, and tight regulations in the long run. In summary, this study provides a comprehensive overview of privacy and security issues in the medical IoT with countermeasures and solutions that can be taken along with the key challenges and future directions. We believe that this will be useful and that it will provide assistance and open new ways for medical practitioners, researchers, academics and students, and other relevant stakeholders who are interested.

**Author Contributions:** Conceptualization, N.N.T., R.K.M., M.E., H.W. and A.W.; methodology, N.N.T., M.E. and A.A.G.; software, N.N.T.; M.I.A. and A.A.G.; validation, N.N.T.; formal analysis, N.N.T. and M.I.A.; investigation, N.N.T. and R.K.M.; resources, N.N.T.; data curation N.N.T., A.A.G.; writing—original draft preparation, N.N.T. and R.K.M.; writing—review and editing, N.N.T. and R.K.M.; visualization, M.E. and M.I.A.; supervision, M.E., A.A.G., H.W. and A.W.; project administration, N.N.T. and R.K.M.; funding acquisition, M.E. and M.I.A. All authors have read and agreed to the published version of the manuscript.

**Funding:** This research received no external funding.

**Institutional Review Board Statement:** Not applicable.

**Informed Consent Statement:** Not applicable.

**Conflicts of Interest:** The authors declare no conflict of interest.

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
