# Peer review of "Security and Privacy Issues in Medical Internet of Things: Overview, Countermeasures, Challenges and Future Directions"

_sustainability, doi:10.3390/su132111645_

Round 1

Reviewer 1 Report

The review presented is good, I recommend this paper. 

Reviewer 2 Report

The authors provide a comprehensive review on security and privacy issues on Medical Internet of Things. The manuscript is well organized and the authors provide a sufficient set of references in order to support their findings. The manuscript fits in the scope of the journal and therefore i suggest publication after minor revision.

My only suggestion is more a presentation type : it will be beneficial to the readers if the authors add in section 7, a table like Table 5 in order to associate the findings with references in a compatible way.

Reviewer 3 Report

This paper presents the study of a security related issues in medical Internet of Things (MIoT). The MIoT concept is now a day of extreme interest, particularly security and privacy concerns, but the work is in an early stage. The paper need some modifications to increase its readability. I have the following comments.

  1. Although the introduction section attempt to follow the typical structure with general description, it should include the significance of the study, motivation, research questions, the research challenges, and contributions. Moreover, the contribution of section 5 should be in section 1; if required, it can be a subsection of 1.
  2. The Related work section needs to be heavily improved. At first, the related-work table is ambiguous; does title is a core parameter for comparing existing work? Secondly, as the authors mentioned the limitations of existing works in line numbers 520-524, so these things should be reflect in the table, which will increase the readability of the paper. I would recommend only reference number and year in the first column of a related work comparison table.
  3. I would recommend presenting the comparison of existing related literature with the proposed work in a tabular form, which will provide an idea to the reader of how the proposed review is novel.
  4. Another important concern is about “countermeasures and solutions” section. The authors must present the solutions regarding specific attacks, as the authors classified the attacks based on the architecture.
  5. As the authors mentioned the weakness of the existing studies in line numbers 508-515, which is the main contribution of this study. Therefore, I strongly recommend that the layered architecture, what type of issues (security or privacy) and its countermeasures are there in existing works should be included in the comparison. Moreover, I suggest that please use a table to compare the findings, which will increase the readability of the manuscript.
  6. Challenges should be classified based on layered architecture, security and privacy issues, and countermeasures.
  7. In the MIoT system, various protocols, including routing, play a crucial role in efficient service provisioning; in this context, if possible, please include such things in the paper.
  8. As the authors mentioned that the existing work lacks deep analysis about the pervasive MIoT environment in line 523; however, I did not see such an environment in the proposed review work. Please, clarify this.
  9. I recommend include future directions in the conclusion section.
  10. More discussion is also expected for section 7.
  11. In terms of references, some of them are outdated.
  12. The quality of the figure should be improved. All the graphs should be reproduced in Matplotlib/Matlab or other similar programme and not in MS Excel.
  13. Please fix the typos.
  14. I would suggest to discuss some recent works:
  • Bhandari, K. S., Ra, I. H., & Cho, G. (2020). Multi-Topology Based QoS-Differentiation in RPL for Internet of Things Applications. IEEE Access.
  • Bhandari, K. S., Seo, C., & Cho, G. H. (2020). Towards Sensor-Cloud Based Efficient Smart Healthcare Monitoring Framework using Machine Learning.

Overall, I do not recommend the current form of the paper for publication, and authors are encouraged to update and significantly improve their paper.

Round 2

Reviewer 3 Report

I consider that the revised manuscript has undergone substantial improvement. The authors have addressed all the comments and suggestions. I would recommend accepting this paper.